# Reduction of Nitrite in Canned Pork through the Application of Black Currant (*Ribes nigrum* L.) Leaves Extract

**DOI:** 10.3390/molecules28041749

**Published:** 2023-02-12

**Authors:** Karolina M. Wójciak, Karolina Ferysiuk, Paulina Kęska, Małgorzata Materska, Barbara Chilczuk, Monika Trząskowska, Marcin Kruk, Danuta Kołożyn-Krajewska, Rubén Domínguez

**Affiliations:** 1Department of Animal Raw Materials Technology, Faculty of Food Science and Biotechnology, University of Life Sciences in Lublin, 20-704 Lublin, Poland; 2Department of Chemistry, Faculty of Food Science and Biotechnology, University of Life Sciences in Lublin, 20-950 Lublin, Poland; 3Department of Food Gastronomy and Food Hygiene, Institute of Human Nutrition Sciences, Warsaw University of Life Sciences SGGW, 02-776 Warsaw, Poland; 4Centro Tecnológico de la Carne de Galicia, Avd. Galicia n° 4, Parque Tecnológico de Galicia, 32900 San Cibrao das Viñas, Ourense, Spain

**Keywords:** canned meat, nitrite-free, antioxidants, lyophilization, *R. nigrum* L., black currant leaves, TBARS

## Abstract

Sodium nitrite is a multifunctional additive commonly used in the meat industry. However, this compound has carcinogenic potential, and its use should be limited. Therefore, in this study the possibility of reducing the amount of sodium(III) nitrite added to canned meat from 100 to 50 mg/kg, while enriching it with freeze-dried blackcurrant leaf extract, was analyzed. The possibility of fortification of canned meat with blackcurrant leaf extract was confirmed. It contained significant amounts of phenolic acids and flavonoid derivatives. These compounds contributed to their antioxidant activity and their ability to inhibit the growth of selected Gram-positive bacteria. In addition, it was observed that among the three different tested doses (50, 100, and 150 mg/kg) of the blackcurrant leaf extract, the addition of the highest dose allowed the preservation of the antioxidant properties of canned meat during 180 days of storage (4 °C). At the end of the storage period, this variant was characterized by antiradical activity against ABTS (at the level of 4.04 mgTrolox/mL) and the highest reducing capacity. The addition of 150 mg/kg of blackcurrant leaf extract caused a reduction in oxidative transformations of fat in meat products during the entire storage period, reaching a level of TBARS almost two times less than in the control sample. In addition, these products were generally characterized by stability (or slight fluctuations) of color parameters and good microbiological quality and did not contain N-nitrosamines.

## 1. Introduction

Oxidation of meat and meat products leads to the deterioration of their nutritional value, color, and sensory properties, as well as the formation of harmful compounds (e.g., aldehydes and ketones) [1,2]. The mechanism of oxidation of lipids, proteins, and pigments has been very well explained by various authors [1,2,3,4,5]. In general, free radicals are considered the main elements responsible for the oxidation of lipids and proteins. Therefore, in the last decades, several strategies to improve the oxidative stability of meat and meat products were studied, including the addition of natural antioxidants [6,7,8], the use of active packaging [9,10], or a combination of both. These substances can inhibit the activity of free radicals and improve the shelf life of the meat products [2,10]. Sodium(III) nitrite is a strong antioxidant that not only protects these products against the negative effects of oxygen, but also renders them with the characteristic color, taste, and flavor [11,12]. In addition, this salt exhibits antimicrobial action against various foodborne pathogens (*Escherichia coli, Clostridium botulinum, Clostridium perfringens,* and *Bacillus cereus*) [11]. For these reasons, sodium(III) nitrite can be considered as a multifunctional additive, and replacing it in meat products’ processing is a major challenge for researchers and professionals in the meat industry [13].

Unfortunately, the latest research points out that nitrite salts pose harmful effects on human health, which was confirmed by the detection of *N*-nitrosocompounds [14]. *N*-nitrosamines (NAs) are mutagenic, genotoxic, and cancerogenic compounds that are formed in cured products during the process of production or storage [15,16]. In addition, NAs could also be produced during the complex digestive process in the human body [15,16]. Hence, there is a search for new methods to protect the safety of meat products. To balance the benefits of sodium(III) nitrite and its adverse effects, it was suggested that the amount of nitrite added to meat products can be reduced. However, to maintain food safety, some additional methods of protection should also be applied [16].

In this sense, some studies have proposed the use of high-nitrate plant extracts or powders, such as those obtained from *Beta vulgaris* [13], radish [17], or cruciferous vegetables [18], to reduce the addition of synthetic nitrite to meat products. Additionally, because secondary plant metabolites are known to have various positive effects on human health, scavenge free radicals, and act as reactive metal chelators, their addition to meat products was also proposed [19,20]. For this purpose, the extracts of various plant materials have been tested: black barberry (*Berberis crataegina* L.) [21]; pomegranate (*Punica granatum* L.) and pistachio (*Pistacia vera* L.) green hull [22]; grape seed and chestnut together with olive pomace hydroxytyrosol [23]; *Caesalpinia sappan* L. [24]; tomato (*Solanum lycopersicum* L.), pomace extract, and peppermint (*Mentha piperita* L.) essential oil [25]; red dragon fruit (*Hylocereus polyrhizus*) peel extract [26]; and bilberry (*Vaccinium myrtillus*) and sea buckthorn (*Hippophae rhamnoides*) leaves [27]. It should be noted that most of the research works focused on cooked pork or beef sausages, while canned meat has not been tested so far.

Black currant (*Ribes nigrum* L.) belongs to the *Grossulariaceae* family, which contains over 150 species. This plant does not require specific conditions for cultivation and can grow in various types of soils [28,29]. Its fruits are rich in different polyphenolic compounds, especially anthocyanins [28,29], while the leaves contain mainly quercetin-3-O-derivatives, myricetin, and kaempferol derivatives, as well as neochlorogenic acid, caffeic acid, gallic acid, chlorogenic acid, and catechins [30,31,32]. However, the content of polyphenols is higher in leaves than in fruits [29,30,31]. The soil type and harvest time determine the antioxidant activity and chemical composition of the extracts obtained from these raw materials [29,30]. Nevertheless, both the leaves and fruits of black currant exhibit various biological effects, such as antimicrobial, antihypertensive, antimutagenic, antidiabetic, analgesic, anti-inflammatory, and anticancer properties, and can inhibit cell proliferation [29,31]. Leaves of black currant plants were used in natural medicine in the form of extracts to treat inflammatory disorders and as a diuretic, while in the form of infusions, they were used to regulate the function of kidneys [30].

Despite the high antioxidant potential of black currant leaves and their possible use as an antioxidant in meat products, studies focusing on this subject are limited. Among them, only in the study of Nowak et al. [33], extracts obtained from the leaves of cherry (*Prunus cerasus* L.) and black currant (*R. nigrum* L.) were analyzed to determine their antimicrobial properties in pork sausages with no nitrite addition.

Our previous research [34] proved that the amount of sodium(III) nitrite added to canned pork can be safely reduced without adverse effects, such as the presence of NA, exceeding the amount of malondialdehyde (MDA), and formation of pathogenic bacteria. However, it was noted that the antioxidant properties of the product containing only 50 mg/kg of sodium(III) nitrite were reduced during storage.

Therefore, the present study aimed to evaluate the possibility of producing canned pork, in which the amount of sodium(III) nitrite added was reduced from 100 mg/kg (following Regulation No. 1333/2008 [35] for canned meat products) to 50 mg/kg, along with simultaneous fortification with water-lyophilized leaf extract of black currant during six months (180 days) of storage. The assessment of the quality of the products was based on the assessment of color-forming properties (L*, a*, b*, and nitrosochemochrome), antioxidant properties (in the ABTS, DPPH, and FRAP tests as well as the TBARS test) and antimicrobial properties, important from the point of view of consumers. N-nitrosamines (NAs) content as a critical point in assessing product safety was also assessed in these studies.

## 2. Results and Discussion

### 2.1. Characteristics of Black Currant Leaf Extract

#### 2.1.1. Chemical Analysis and Antioxidant Capacity

The lyophilized aqueous extract of black currant leaves (Figure 1) was characterized by a slightly acicular consistency, mild golden color, and delicate fruity aroma. The extract was easily soluble in water.

The extraction yield was satisfactory and amounted to over 18%. Chemical analysis revealed the complex chemical composition of the prepared extract (Table 1). The antiradical activity of the tested extract, determined in the ABTS radical model system and expressed as the EC50 value (11.1 ± 0.06 µg/mL), was significant (*p* < 0.05). The EC50 value was even lower than of DPPH (32.5 ± 0.21 µg/mL). The active compounds present in the prepared extract were phenolic acids and derivatives of flavonoids with the concentration of acids found to be dominant over that of flavonoids. This was evidenced by the results of both spectrophotometric analyses and the detailed analysis of the phenolic compound profile using the HPLC method (Table 1 and Figure 2). Of the ten compounds identified in the extract, half were phenolic acids and the other half were glycosidic derivatives of quercetin, luteolin, and apigenin (Figure 2). However, in terms of quantity, the content of phenolic acids, calculated in accordance with the HPLC results, exceeded 99% (Table 1). The chemical content of the investigated dry black currant leaf extract differed from that reported by other authors both quantitatively and qualitatively. The highest concentration of chlorogenic acid was recorded in the tested extract, while in other studies the highest concentration was noted as quercetin-3-*O*-glucoside [30]. This may be due to differences in the methods used for extract preparation as well as analysis.

#### 2.1.2. Antimicrobial Activity of Black Currant Leaf Extract

MIC refers to the lowest concentration of the antimicrobial agent that completely inhibits the visible growth of the tested microorganisms [36]. As shown in Table 2, depending on the species, the tested strain displayed different levels of susceptibility to the extract. The black currant leaf extract mostly inhibited the growth of Gram-positive bacteria. However, no growth inhibitory effect toward LAB, *E. coli, Enterococcus faecalis, Clostridium sporogenes*, and *Salmonella Hofit* IFM 2318 was observed. This lack of suppressive effect on LAB is advantageous as it may allow the use of the extract in fermented meat products. For *C. sporogenes*, a reduction in the size of bacterial colonies was noted with an increase in the amount of extract in the culture medium. This suggests that a higher concentration of black currant leaf extract may be needed to inhibit the growth of this species. The results of our study agree with those reported in the works of Staszowska-Karkut et al. [30] and Paunović et al. [29], which demonstrated the ability of black currant leaf extract to limit the growth of microorganisms. However, the likelihood of bacterial growth inhibition was influenced by the strain properties and the method used for MIC determination as well as for the extraction of black currant leaves.

### 2.2. Characterization of Canned Pork Containing Black Currant Leaf Extract

#### 2.2.1. Color Parameters and Nitrosohemochrome

The color properties of canned pork samples with leaf extract added are shown in Table 3. In general, the effect of time on the lightness of the products was not confirmed. All products were characterized by the stability of the L* parameter during refrigerated storage, except for sample B100 for which a significant increase in lightness (*p* < 0.05) was noted on the 180th day of the analysis. For this sample, the L* parameter was on average higher by almost 16 units compared to the other test variants at that time. Taking into account the effect of the dose of lyophilized extract on the lightness of meat products, no significant differences were found between the samples with a lower dose of the extract (B50) and the control sample (C). On the other hand, for sample B100 and B150, a significant (*p* < 0.05) increase in the lightness value was noted after 180 and 90 days of storage, respectively. Moreover, during two months of storage, the a* parameter (redness) was stable for samples B100 and B150 and the values were similar between the samples. Changes were observed after the next 60 days in the samples containing 100 mg/kg of leaf extract, with a slight but statistically significant (*p* < 0.05) increase in a* value. An opposite trend was noted for changes in the a* value in the B150 sample. A gradual increase in the value of the b* (yellowness) parameter was noted in samples B50 and B150 (from 7.39 to 9.18 and from 8.08 to 8.89, respectively). However, a significant (*p* < 0.05) increase in the value of this parameter was noted in sample B100 (from 8.37 to 13.31, difference: approximately 4.94). Moreover, it was observed that this particular sample was highly unstable during the six months of storage in comparison to other samples with respect to both redness and yellowness parameters.

During 180 days of storage, samples with lyophilized black currant leaf extract at the level of 50 mg/kg (B50) and the batches with the highest amount of extract (B150) contained a stable amount of nitrosohemochrome (NO-Mb), and only a slight but significant (*p* < 0.05) decrease in content was observed during the storage time, compared to sample B100, in which sharp, alternating increases and decreases in the amount of NO-Mb were noted. However, at the end of the storage period, all the samples had a similar amount of this pigment (about 22.38 mg/kg). The protein responsible for the color of meat is myoglobin (Mb). Mb is influenced by the state of its heme group, which contains an iron ion. Based on the content of Mb, meat products can appear dull brown or purplish red. The addition of nitrite leads to the formation of nitrosylmyoglobin, which is later stabilized to nitrosyl hemochrome upon thermal treatment, and as a result meat products acquire the characteristic pink color [12,37]. It should be mentioned that the iron ion from the heme group acts as a double agent—it contributes to color formation and also acts as a pro-oxidant due to its ability to donate electrons with ease [38]. Suman and Joseph [37] reported that lipid oxidation products (e.g., aldehydes) can also affect the color of meat products by destabilizing the heme group. The addition of antioxidant substances (natural or synthetic) causes the chelation of Fe^2+^ ions [38]. Riviera et al. [12] pointed out that for producing meat products with stable and acceptable color for commercial purposes, about 10–15 ppm of NaNO_2_ should be added, while Food Chain Evaluation Consortium [16] suggests an amount of 55–70 mg/kg (for nontraditional products).

Bae et al. [39] investigated the effect of alternating cured process on, among others, the number of pigments in pork products. They observed no significant differences between the control (0.1% of NaNO_2_) and test samples with radish or celery powder (0.15% and 0.30%). In general, the redness of all samples varied from 8.30 to 8.55 (control), but the values of b* parameter were higher in products containing celery powder at an amount of 0.30% (b* = 8.66). Bae et al. [39] also noted that the addition of radish powder and a long incubation time resulted in higher yellowness in the samples. The authors suspected that pigments presented in celery could affect the value of the b* parameter. In addition, they observed that incubation time affected the amount of nitrosyl hemochrome to a greater extent than the quantity of the added powder. This is explained by the fact that a higher volume of nitrite is converted by bacteria from a vegetable source. In our study, low-nitrate plant material was used, which may explain the low value of NO-Mb (<30 mg/kg in comparison to the value reported by Bae et al. [39]). It could also be assumed that 50 mg/kg of sodium(III) nitrite was not enough to support a higher amount of nitrosyl hemochrome but was high enough to result in an appropriate value of the a* parameter (approximately 10 throughout the storage period in comparison to the value of about 8 after production in Bae et al.’s study). Data obtained from our study show the influence of black currant leaf extract on the yellowness of canned meat—the value of the b* parameter is higher than that determined by Bae et al. [39]. Moreover, Nowak et al. [33] observed an increase of the b* parameter during storage in pork sausages containing cherry and black currant leaf extract. The authors concluded that this could be related to the color of plant extracts. Similarly, Sun and Xiong [40] noted that the amount of NO-Mb was not directly proportional to the redness of beef patties (samples containing pea protein isolate and pea protein hydrolysate had a lower value of nitrosyl hemochrome compared to their a* values). In addition, the authors assumed that other pigments may also influence the color of beef patties. Moreover, it could be possible that nitrogen groups from pea proteins may bind heme iron and, therefore, decrease the level of nitrosyl hemochrome. 

#### 2.2.2. Antioxidant Abilities

##### Antiradical Activity and FRAP

The results obtained in the analysis of the antioxidant abilities (ABTS^+^, DPPH, FRAP) of canned pork samples are presented in Table 4. There were no differences in the ability to neutralize synthetic radicals (ABTS^+^ or DPPH) as well as in the FRAP between the control sample (C) and the B50 sample during refrigerated storage. However, referring to the ABTS test, after 1 day of storage, the free radical scavenging capacity was high in samples B50 and C (3.95 and 3.90, respectively, *p* < 0.05). On the other hand, immediately after production (1 day) there was no effect of the extract dose on the antiradical effect against DPPH. Although the present study confirmed the antioxidant effect of extracts in model systems, immediately after the production of canned meat, no effect of the amount of plant extract addition on the ability to neutralize free radicals was observed, both in the ABTS and DPPH tests. The effect of the antioxidant compound in the food matrix may differ significantly in activity from the purified extract. This is influenced by many factors, including matrix properties (pH, water activity), antioxidant structure, and the resulting structure–activity relationship (SAR). In addition, it is well known that antioxidants can be significantly altered by processing. In particular, exposure to high temperatures, such as pasteurization [41] or sterilization [42], may be the main cause of the reduction of the natural amounts/properties of plant antioxidants. Many plant secondary metabolites act as antioxidants and pro-oxidants and can affect the concentration of reactive oxygen species depending on the reaction conditions. Under certain conditions, flavonoids and phenolic acids have the ability to act as pro-oxidants, and their activity depends on the chemical structure and concentration of the compound and the environment of their reaction, such as pH, the presence of metals (Cu and Fe), and temperature. This may explain the lower antioxidant activity of products containing the addition of plant extract (containing, among others, polyphenols) immediately after production, in that the greater the addition of the extract, the lower the effectiveness against ABTS radicals (Table 4). In addition, the heating process also disrupts muscle cell structure, inactivates antioxidant enzymes, and produces myoglobin catalytic iron, leading to an intense pro-oxidative environment, which may have affected the antioxidant activity of canned meat in the early post-production period. With the increase in storage time, the antiradical effect of extracts from meat products increased and was higher in samples with a higher content of extracts from currant leaves. Notably, this increase was more intense in the test with ABTS than with DPPH, which was probably related to the low antiradical activity of hydrophobic compounds present in the products. The antioxidant properties of sample B50 were lower than sample B100 and B150 with the ABTS test. At the end of the storage period, samples with the lowest amount of extract addition showed a similar value, while canned pork with 150 mg/kg of black currant leaves extract presented the strongest antioxidant properties (4.04 mgTrolox/mL). However, statistical analysis did not show any significant difference in the values. Conversely, on the 60th day of the analysis, variants with higher doses of lyophilized extract obtained from black currant leaves (B100 and B150) were characterized by a higher antiradical capacity than the remaining variants, although it decreased over the time of the analysis (Table 4). Khaleghi et al. [21] observed an increase in antioxidant properties (DPPH method) in beef sausages with an increase in the amount of black barberry extract added; however, when sodium(III) nitrite was also added at an amount of 90 mg/kg, the DPPH value was lower. Seo et al. [24] noted higher DPPH scavenging ability in the sample with 0.1% *C. sappan* extract than the sample with 0.05% of extract and 0.004% of nitrite. In addition, it can be postulated that in addition to the added antioxidants present in the extracts, other substances, such as proteins present in the meat samples, may affect the antioxidant activity of the samples. On the one hand, the presence of myoglobin, which is easily oxidized, may be associated with increased antioxidant activity in the DPPH test [43]. This can be explained by the fact that the oxidation of meat leads to the formation of compounds that act as scavengers of free radicals that could not be detected by the TBA test, which measures the outcome of fat oxidation [44]. On the other hand, meat whites may be a source of biologically active peptides, and their release from the protein chain may increase the antiradical effect of canned meat extracts. In this context, the addition of natural polyphenolic compounds may promote this bioactive role of peptides [45].

Metals, such as copper and iron, can act as pro-oxidants, which is related to the Fenton reaction and change of their oxidation form. Therefore, it was assumed that metals can determine the oxidation stability of meat products [2]. The ability of polyphenols to donate electrons demonstrates the reducing power of these biomolecules and is also representative of their antioxidant activity. As observed in this study, a systematic increase (from 1.84 to 2.00) in iron ion-reducing power was noted in sample B150, with the highest amount of black currant leaf extract (150 mg/kg) (Table 4). Samples C (control), B50, and B100 presented similar values of this parameter (1.67, 1.68, and 1.64, respectively) at the beginning of the storage period. After two months, the values were reduced in C and B50 samples compared to samples B100. Subsequently, a gradual increase in the values of the FRAP parameter was noted in C and B50 samples, while the sample with 100 mg/kg of black currant leaves extract was stable. However, at the end of the storage period, both samples showed similar values (1.74 and 1.73, respectively). Mira et al. [46] investigated the ability of selected flavonoids to reduce copper and iron and concluded that they depend both on the standard metal redox potential (E0) as well as on the flavonoid structure. Changes in both of these parameters may have contributed to the observed results in the FRAP antioxidant test.

##### Secondary Lipid Oxidation Products 

The results obtained in the analysis of the secondary lipid oxidation products were detected (Figure 3). Determination of the MDA content is one of the most commonly used methods for evaluating the degree of lipid rancidity. During lipid oxidation, various hazardous products are produced, among which aldehydes are highly toxic as they are characterized by mutagenic, cytotoxic, and pro-inflammatory properties [2]. As presented in Figure 3, from the first day of storage (4 °C) there was a significantly (*p* < 0.05) higher MDA content in the sample with only 50 mg/kg of extract compared to other variants, although the B50 sample had the same TBARS level as the control sample throughout the whole period up to 180 days. After 90 days of storage, a rapid increase in the amount of MDA was observed in samples B50 and B100 (approximately 0.034 and 0.021, respectively) compared to sample B150. The addition of 100 and 150 mg/kg black currant leaf extract resulted in lower (*p* < 0.05) values of the TBARS parameter compared with the C and B50 sample. However, the TBARS values were stable during six months of storage only for sample B150. In fact, similar values were described in the B150 samples between 0 and 180 days (0.033 and 0.029 mg MDA/kg in 0 and 180 days, respectively). Additionally, the significant decrease of the MDA content between 90 and 180 days in the B50 and B100 samples is expected since the secondary lipid oxidation products could be further degraded to other more stable compounds [2]. The acceptable amount of MDA in meat products is around 2–2.5 mg/kg [2]. Other authors conclude that in some meat products, this limit could be 0.6 mg MDA/kg [47,48]. In our study, none of the canned pork samples showed MDA levels exceeding this limit since very low values were observed in all samples (between 0.018 and 0.047 mg MDA/kg) but sample B50 after 90 days of storage (0.07 mg MDA/kg). A possible explanation for these low TBARS values could be related to the limited exposition of canned meat to oxygen [47]. It is well known that oxygen plays a vital role in lipid oxidation, both in the initiation and propagation phases [2]. Thus, canned meat in oxygen-free packaging determines the low lipid oxidation. 

Natural substances with antioxidant properties are added to meat products to prevent oxidation. In this sense, the antioxidant capacity of the extract from black currant leaves could be another explanation for the limited lipid oxidation observed in our samples. However, the use of antioxidants in combination does not always give positive results. During 30 days of storage (4 °C) of beef sausages with the addition of nitrite and black barberry extract (30, 60, and 90 mg/kg), a negative interaction was observed between the two additives at the highest concentrations. The synergistic effect was observed when 90 mg/kg extract was used in combination with 30 mg/kg sodium(III) nitrite [21]. However, different results were observed in the study of Šojić et al. [25], in which a combination of sodium nitrite, tomato pomace extract, and peppermint essential oil allowed for the obtaining of much lower levels of MDA in cooked pork sausages (60 days of storage at 4 °C) than samples containing nitrite and a single plant additive. A similar observation was made by Seo et al. [24]. They observed that samples with *C. sappan* (separately and with nitrite) had a similar amount of MDA as samples containing only sodium(III) nitrite (0.007%) during one month of storage (4 °C). However, the authors noted that a combination of two antioxidants allowed for the obtaining of slightly better results. As mentioned earlier, the sample containing both antioxidants showed greater antioxidant capacity (DPPH method) than other samples, but this did not translate into better MDA inhibition [24]. A similar finding was also noted in our study (Table 1). Although, Mäkinen et al. [27] found higher antioxidant properties in pork sausages containing bilberry and sea buckthorn leaf extract in a higher concentration than the control sample (nitrite and ascorbic acid) during 20 days of storage. The addition of 2.0% and 1.0% of bilberry leaves and 0.2% of sea buckthorn leaves resulted in the inhibition against lipid oxidation. 

#### 2.2.3. Consumer Safety

NAs (i.e., NDBA, NDMA, NDEA, NDPA, NMOR, NPIP, and NPYR) were not detected after 1 and 180 days of storage in the samples (data not presented). These compounds are formed from nitrosating derivatives (e.g., NO) and secondary amines—a combination that allows the formation of the most stable NA [49]. During the reaction of nitrite with the heme group or free thiol groups, nitric oxide is released [50]. NA can be formed in an acidic environment (stomach) or in the presence of microorganisms (large intestine). They are also formed in the product itself during production or while a meal is prepared at home (when the applied temperature is higher than 100 °C and in the presence of fat or during reheating) [49,50]. Endogenous NA formation in the colon is associated with the nitrosylation of heme (resulting from nonabsorbed red meat residues) and products formed from the metabolism of amino acids by microorganisms [51].

De Mey et al. [52] pointed out that following good manufacturing practices are necessary to prevent the formation of NA—storage of meat for a longer period may increase the chance of NA formation, while spices containing pyrrolidine or pyrroperine may act as NA precursors. Moreover, Park et al. [51] conducted extensive research on the presence of NA in various types of food (including milk and milk products, seafood, alcoholic beverages, oils, meat, and processed meat). The authors detected *N*-nitrosodimethylamine (NDMA) in all the examined processed meat products and *N*-nitrosodiethylamine (NDEA) and N-nitrosopiperidine in most of the tested products. According to Özbay and Şireli [53], NDMA can lead to diarrhea, abdominal cramps, vomiting, headache, as well as carcinogenic and mutagenic effects. NDEA also has carcinogenic effect on the esophagus and liver, while *N*-nitrosodipropylamine exhibits a carcinogenic effect on the lungs and esophagus. The addition of antioxidant substances (e.g., ascorbic acid or tocopherol) may inhibit the formation of NA [53]. 

After the first and last days of storage, no pathogenic microorganisms (i.e., *Clostridium perfringens, Listeria monocytogenes,* or *Salmonella*) were detected in the tested samples (data not presented). Paunović et al. [29] tested the effect of extracts from fruits and leaves of black currant obtained from various types of soils on selected microorganisms (such as *Staphylococcus aureus, Micrococcus lysodeikticus, Bacillus mycoides, Klebsiella pneumoniae, Pseudomonas glycinea, E. coli, Candida albicans, Fusarium oxysporum, Penicillium canescens,* and *Aspergillus glaucus*).

The MIC determined for the extract from fruits was 38.2–170.1 µg/mL, while for the extract from leaves it was 123.0–389.2 µg/mL. Raudsepp et al. [32] compared the ethanolic (20% and 96%) extracts obtained from the fruits and leaves of rhubarb, blue honeysuckle, and black currant for, among others, antimicrobial properties. They found that black currant extract (in all dilutions) inhibited the growth of *Campylobacter jejuni, Salmonellae Enteritidis, E. coli, L. monocytogenes* (extraction with 96% of ethanol), and *B. cereus* (20% and 96% ethanol). The extract from fruits showed slightly lower antimicrobial effects on Gram-positive bacteria; in general, less diluted extracts presented stronger inhibitory properties against microorganisms. However, the extracts from fruits and leaves of black currant did not present strong antimicrobial properties in comparison to rhubarb extracts. Furthermore, Majou and Christieans [54] pointed out that the antimicrobial properties of nitrite depend on various factors (nitrite concentration, pH value, temperature, and presence of curing accelerator). They showed that chemical transformations lead to the formation of peroxynitrite from sodium(III) nitrite, which is suspected to be responsible for antimicrobial properties.

## 3. Materials and Methods

### 3.1. Black Currant Leaf Extract Preparation 

The plant material consisted of black currant (*R. nigrum* L.) leaves, which were collected in May before the flowering of the bushes. The extraction conditions were established in our previous studies [55,56,57]. The leaves were initially dried in a shaded, airy place and then in a dryer at 60 °C. Dry leaves were used for preparing water extract by ultrasound-assisted extraction (10 min) with hot distilled water (90 °C) at a plant-to-solvent ratio of 1:10 (*m*/*v*). The infusions were placed in an ultrasonic bath for extraction using Sonic 6D equipment (Polsonic Palczynki Sp. J., Warsaw, Poland). The ultrasound frequency was set as 40 kHz and sound intensity as 320 W/cm^2^, temperature as 30 °C. The obtained infusions were filtered after 30 min, cooled, frozen (−18 °C), and then lyophilized. Freeze drying was carried out for 72 h using a freeze dryer (Free Zone 12 lyophilizer, Labconco Corporation, Kansas City, MO, USA) at −80 °C and 0.04 mbar. The lyophilized dry extracts were stored in airtight plastic containers, protected from light, at room temperature until analysis. 

### 3.2. Chemical Analysis and Antioxidant Capacity of Black Currant Leaf Extract 

#### 3.2.1. Total Phenolic Content

The total phenolic content of the extracts was analyzed by the Folin–Ciocalteu (FC) method using gallic acid (25–500 mg/L) as standard [58]. Briefly, 0.06 mL of extract was mixed with 0.54 mL of distilled water, 1.5 mL of FC reagent (diluted 1:10 with distilled water), and 1.2 mL of sodium bicarbonate solution (7.5%). The samples were incubated in the dark at room temperature for 30 min and then the absorbance was measured (750 nm) on a Cary 50 spectrophotometer (Varian, Palo Alto, CA, USA). The results expressed as mg gallic acid equivalent/g dry extract.

#### 3.2.2. Total Flavonoids

Total flavonoids of the extracts were determined using a spectrophotometric method based on the formation of a colored complex between the flavonoid and AlCl_3_ [59]. Briefly, the lyophilized extract was dissolved in water to prepare a solution with a starting concentration of 1 mg/mL. Then, 0.25 mL of the prepared solution was mixed with 0.75 mL of ethanol (96%), 0.05 mL of AlCl_3_ (10%), 0.05 mL of sodium acetate (1 M), and 1.4 mL of distilled water. The absorbance of the resulting solution was measured at 415 nm. Total flavonoids were expressed as quercetin equivalent based on a calibration curve previously prepared for this compound and were presented in mg quercetin/g dry extract. 

#### 3.2.3. Dihydroxycinnamic Acids

Samples of dissolved extracts were prepared as described for the analysis of polyphenolic compounds. The content of total hydroxycinnamic acid was expressed as chlorogenic acid equivalent, as described by Nicolle et al. [60]. Briefly, 1 mL of the extract was added to 2 mL of 0.5 M HCl, 2 mL of Arnow’s reagent (10 g of sodium nitrite and 10 g of sodium molybdate, made up to 100 mL with distilled water), 2 mL of NaOH (at a concentration of 2.125 M), and 3 mL of water. Each of the prepared solutions was compared with the corresponding mixture that did not contain Arnow’s reagent. The absorbance was read at 525 nm.

#### 3.2.4. HPLC Analysis

The high-performance liquid chromatography (HPLC) method was used for a detailed investigation of changes in the lyophilized water extracts obtained from the leaves of black currant. The analysis was performed using an Empower-Pro chromatograph (Waters), equipped with a quaternary pump (M2998 Waters) with a degasser and a UV–VIS diode array detection system. Separation was performed on a column filled with modified silica gel RP-18 (Atlantis T3, Waters; 3 µm, 4.6 × 150 mm). The mobile phase consisted of A (1% acetic acid) and B (acetonitrile), in which the concentration of solvent B changed as follows: until 0–8th min, 8–12%; in the 10th min, 20%; and in the 25th min, 25%; the flow speed was set at 1 mL min^−1^. Detection was carried out at 330 nm. The identified compounds (marked by peak numbers on the chromatogram) were quantified according to the calibration curves prepared for each compound, and their content was expressed as mg/g dry extract.

#### 3.2.5. Vitamin C Content

The content of vitamin C was determined with the spectrofluorimetric method described by Wu [61] with necessary modifications. Compounds including 3% metaphosphoric acid, 7 M HCl, 0.1 M Na_2_S_2_O_3_, and 0.005 M H_2_SO_4_ were used in the analysis. An oxidizing solution was prepared by dissolving 1.3 g I_2_ in 10 mL of 40% KI, in which 0.1 mL of 7 M HCl and distilled water were added to a volume of 100 mL. The 0.1 M Na_2_S_2_O_3_ solution was made by dissolving 1.25 g of the reagent and 0.01 g Na_2_CO_3_ in 50 mL of water. The derivatization reagent was prepared using 10 mg o-phenylenediamine (OPDA) dissolved in 10 mL of 0.005 M H_2_SO_4_. 

The lyophilized extracts of black currant leaves (0.2 g) were diluted in 10 mL of 3% metaphosphoric acid. To the sample extract (2 mL) or standard (L-ascorbic acid, 50–500 µg/L), 0.3 mL portions of a 0.005 M solution of iodine in potassium iodide were added. The solution was vortexed for 1 min and supplemented with 0.3 mL portions of 0.01 M Na_2_S_2_O_3_. The pH of the sample was adjusted to approximately 6.0 by adding 0.3 mL of 2 M NaOH, and derivatization was carried out by adding 0.3 mL portions of the OPDA solution. The solution was stirred for 30 min at the maximum stirring force. Then, it was diluted to 50 mL with distilled deionized water. The determinations were conducted on a Cary Eclipse spectrofluorometer (Varian, Palo Alto, CA, USA) at an excitation wavelength of λ = 365 nm and an emission wavelength of λ = 425 nm. The content of vitamin C was calculated based on a standard curve prepared using an aqueous solution of l-ascorbic acid standard and expressed as l-ascorbic acid (µg)/100 g sample (dry basis). 

#### 3.2.6. Antioxidant Capacity

The lyophilized extract was dissolved in water to obtain a solution with a starting concentration of 1 mg/mL. The solution was then diluted to produce a series of samples with concentrations ranging from 0.1 to 1 mg/mL. The ability of the samples to scavenge ABTS radical cation (ABTS^•+^) [2,2-azinobis(3-ethyl-benzothiazoline-6-sulfonate)] was determined as described by Re et al. [62]. For this purpose, the working solution of ABTS^•+^ radical was prepared by mixing ABTS (7 mM) with potassium persulfate (2.45 mM). The solution was kept in the dark at room temperature for 18 h and then diluted with 95% ethanol until an absorbance of 0.70 (±0.02) was reached at 734 nm. Next, the sample (20 μL) and the ABTS^•+^ solution (3 mL) were mixed and incubated at room temperature for 10 min. The absorbance of the mixture was measured at 734 nm on a Cary 50 spectrophotometer (Varian, Palo Alto, CA, USA). The control was prepared by adding 20 μL of methanol to ABTS^•+^ solution instead of the sample. In addition to ABTS’s radical scavenging ability, the antiradical capacity of the samples against the DPPH (2,2-diphenyl-1-picrylhydrazyl) radical was measured [63]. Briefly, 0.1 mL of the extract solution was mixed with a freshly prepared methanolic solution of DPPH (0.1 mM, 4 mL). The resulting mixture was vortexed and incubated in the dark at room temperature for 30 min. The absorbance was measured at 517 nm on a Cary 50 spectrophotometer (Varian, Palo Alto, CA, USA). A control was prepared by adding 4 mL of 0.1 mM DPPH to 0.1 mL of methanol (analytical grade) instead of the sample. 

The percentage of absorbance reduction in comparison to the initial value was determined to calculate the inhibition percentage, according to the formula:(1)$$\%AA=\left[1-\left(A_{p}-A_{b}\right)\right]\times 100\%$$
where *AA* is the antioxidant activity of the analyzed sample, *A_p_* is the absorbance of the analyzed sample, and *A_b_* is the absorbance of the blank sample.

Based on the dependence between the antiradical activity of the samples and their concentration (f(c) = %*AA*), the EC50 values were calculated. 

### 3.3. Antimicrobial Activity of Black Currant Leaf Extract

The minimum inhibitory concentration (MIC) of black currant leaf extract was determined according to the recommendations of EUCAST [36]. Mueller–Hinton (MH) agar (Bio-Rad, Hercules, CA, USA) was used as a nonselective medium for the growth of tested microorganisms (Table 2). For lactic acid bacteria (LAB), 10 g/L glucose (Sigma-Aldrich, Poznań, Poland) was added to the MH medium. The bacterial suspensions were prepared using overnight cultures. Aqueous black currant leaf extract was diluted with a molten MH medium to prepare tested concentrations (1–5 mg/mL). After solidification of the MH agar, the tested strain with the adjusted density of 10^4^ colony-forming units (cfu)/mL was spread on the medium. Then, the samples were incubated at 37 °C for 24 h. The positive control consisted of MH agar inoculated with the test bacteria without the extract, while uninoculated plates containing black currant leaf extract served as the negative control. When the visible growth inhibition was observed (judged by the naked eye), regardless of the appearance of a single colony or a thin haze, the MIC of the extract was determined.

### 3.4. Canned Meat Preparation

Canned meat was prepared using pork shoulder and pork dewlap (80%:20%), salt (2%), water (5%), a reduced amount of sodium(III) nitrite (50 mg/kg of meat), and lyophilized black currant leaf extract (0 for control (C)) and 50, 100, and 150 mg/kg of meat for tested batches (B50, B100, and B150, respectively). Meat was purchased from an organic farm (Zakład Mięsny Wasąg SP. J., Hedwiżyn, Poland, organic certificate no. PL-EKO-093027/18). After initial and final grinding (universal machine KU2-3E, Mesko-AGD, Skarżysko-Kamienna, Poland), the material was divided into three variants and subjected to mixing (4–5 min/variant; universal machine KU2-3E, Mesko-AGD, Poland). Then, the meat stuffing was transferred to metal cans (meat filling: 250 g) and sterilized on a vertical steam sterilizer (TYP-AS2, Poland) at 121 °C. After sterilization, the cans were cooled in water and stored (at 4 °C) for further analysis. The experimental canned pork samples were heated at 121 °C, assuming that their degree of heating was achieved as measured with the sterilization value (*F* ≈ 4 min) determined from the formula:(2)$$F=\int_{0}^{1}Ldt$$
(3)$$L=10\frac{T-T_{0}}{z}$$
where *F* is the sterilization value, *L* is the lethality degree, *T*_0_ is the reference temperature (121 °C), and *z* is the sterilization effect factor (10 °C).

The sterilization values were calculated by determining the degree of lethality by measuring temperature every minute. The limits of integration were assumed from 90 °C during the growth phase to 90 °C during the decrease (i.e., cooling). The degree of heating was determined for the cans in their critical zone using an electric thermometer equipped with a thermoelectric sensor. After sterilization, the products were cooled in water and stored. Then, they were divided into four groups, and each group was tested (according to the analysis listed below) immediately after production (day 1), and after 60, 90, and 180 days of storage. The experiment included one-time preparation of 48 (+5 inventory) canned pork samples (12 cans from each test variant: C, B50, B100, and B150). In each study period (1, 60, 90, and 180 days) 3 cans of each variant were tested. The experiment thus planned was repeated three times at about two-week time intervals.

### 3.5. Canned Meat Analysis

#### 3.5.1. Color Parameters (CIE L*a*b*) and No-Mb

The color of the canned meat samples was measured using an X-Rite Color 8200 spectrophotometer (X-Rite Inc., Grand Rapids, MI, USA; port size: 13 mm; standard observer: 10°; illuminant: D65). The samples were cut into cuboids and analyzed at three points [64]. For the determination of nitrosohemochrome, 5 g of samples were homogenized for 1 min in acid acetone. After 30 min of storage in the dark, the samples were centrifuged (4000 rpm for 10 min), and the absorbance was measured (540 nm) using a UV–VIS spectrophotometer (HITACHI U-5100). The results were obtained by multiplying absorbance by 290 and expressed as mg/kg [65]. 

#### 3.5.2. Antioxidant Properties

##### Antiradical Properties and FRAP

Sample Preparation

For the analysis of antioxidant properties, samples were prepared as described by Jung et al. [66] with some modifications. Briefly, 10 mL of ethanol was mixed with 5 g of minced canned meat and homogenized (1000 rpm). Then, the samples were centrifuged (10,000 rpm for 20 min) and filtered. Absorbance was measured using a UV–VIS spectrophotometer (HITACHI U-5100). 

ABTS^•+^ and DPPH

Free radical scavenging activity was determined as described by Jung et al. [66] with some modifications. In the ABTS^•+^ method, 12 μL of supernatant was added to 1.8 mL of diluted ABTS^•+^ solution. Absorbance was measured at 734 nm after 3 min. In the DPPH method, absorbance was measured after 3 min at 517 nm using a diluted DPPH solution with an absorbance of 0.9 ± 0.02. In both these methods, the antioxidant capacity of the samples was calculated from a standard curve (concentration: 0.025–0 mg/mL for DPPH; 15–0 mg/mL for ABTS) and expressed as mg Trolox equivalent/mL.

FRAP

FRAP (ferric ion-reducing antioxidant power) of the samples was measured at 700 nm as described by Oyaizu [67]. The results were expressed as absorbance.

##### Secondary Lipid Oxidation Products

TBARS (thiobarbituric acid reactive substance) parameter was determined as described by Fan et al. [68]. Briefly, the meat samples (10 g) were homogenized using a mixture of ethylenediaminetetraacetic acid (0.1%) and trichloroacetic acid (7.5%). After shaking (30 min) and filtration, the samples were boiled at 100 °C for 40 min, cooled, and centrifuged. Then, chloroform was added, and the samples were manually shaken. After the solutions became clear, measurement was carried out at two wavelengths (532 and 600 nm). The results were expressed as malondialdehyde (MDA) mg/kg according to the formula:(4)$$\mathrm{TBARS}=\frac{\left(A_{532}-A_{600}\right)}{\left(155\times \frac{1}{10}\times 72.6\right)}\times 1000$$
where *A* refers to the absorbance measured at 532 and 600 nm.

### 3.6. Consumer Safety

#### 3.6.1. *N*-Nitrosoamines (NAs) Content

The volatile NAs (VNAs) in the samples were analyzed using the method of Drabik and Markiewicz [69] and DeMey [52]. Briefly, meat samples (50 g) were mixed with 200 mL of 3 N KOH and then VNAs were extracted by vacuum distillation (Heidolph Laborota 4010-digital, Schwabach, Germany). After distillation, 4 mL of 37% HCl was added, and the distillate was extracted three times with 50 mL of dichloromethane. Subsequently, the obtained extract was concentrated. The detection and quantification of selected NAs (*N*-nitrosodibutylamine (NDBA), *N*-nitrosodimethylamine (NDMA), *N*-nitrosodimethylamine (NDEA), *N*-nitrosodipropylamine (NDPA), *N*-nitrosomorpholine (NMOR), *N*-nitrosopiperidine (NPIP), and *N*-nitrosopyrrolidine (NPYR), µg/kg) was performed using a gas chromatograph coupled to a thermal energy analyzer (TEA; Thermo Electron Cooperation). For this, the extracts (5 μL) were injected on a packed column, and chromatographic separation was carried out using argon as carrier gas (25 mL/min). The injection port was set at 175 °C, and the oven temperature was increased from 110 °C to 180 °C at 5 °C/min. The interface and pyrolizer of the TEA were set at 250 °C and 500 °C, respectively. The content of NA in the samples was estimated after 1 and 180 days of storage.

#### 3.6.2. Number of Selected Microorganisms

The microbial count was calculated after 1 and 180 days of storage. The C. perfringens were measured according to [70], Listeria monocytogenes according to [71], and Salmonella according to [72]. For preparing the appropriable dilutions, 180 mL of peptone water was homogenized with 20 g of sample (Stomacher Lab-Blender 400, Seward Medical, London, UK). The results are expressed as cfu/g products.

### 3.7. Statistical Analysis

A two-way analysis-of-variance (ANOVA) model was used for analysis. It included the main effects of the level of extract (0, 50, 100, and 150 mg/kg) and the storage period (1, 60, 90, and 180 days) as well as their interactions. All measurements were carried out in triplicate. The results were analyzed statistically using STATISTICA^®^ 13.1 statistical package (StatSoft) and presented as mean ± standard error using T-Tukey’s range test. 

## 4. Conclusions

The present study evaluated the possibility of reducing sodium(III) nitrite in canned meat, together with simultaneous fortification with lyophilized leaf extract of black currant leaves and its influence on the antioxidant stability of the product during 180 days of storage. The analyses of free radical scavenging ability (ABTS^•+^, DPPH) and iron ion-reducing potential allowed us to confirm the high antioxidant properties of meat products with black currant leaf extract. These results were further confirmed by the analysis of secondary lipid oxidation products, which showed low amounts of MDA in the tested products. In addition, no negative, pro-oxidative interactions were noted between sodium(III) nitrite and black currant leaves extract, even in the sample with the highest (150 mg/kg) amount of added extract. Moreover, no NAs were detected after production in any of the samples. The addition of black currant leaf extract at an amount of 150 mg/kg of meat stuffing can be recommended.

Our research confirms the possibility of reducing the amount of sodium(III) nitrite in canned meat and at the same time maintaining the safety and quality of the product.

## Figures and Tables

**Figure 1 molecules-28-01749-f001:**
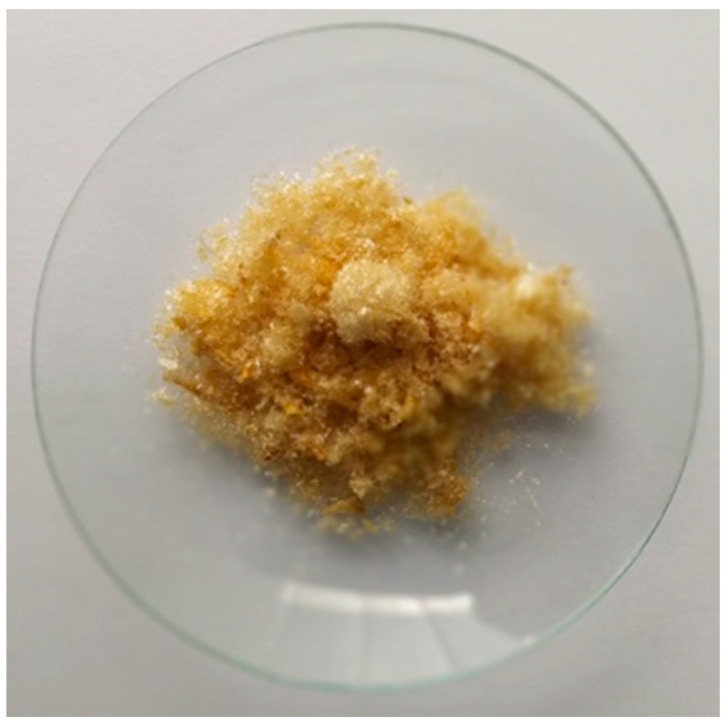
Water extract of black currant leaves after lyophilization.

**Figure 2 molecules-28-01749-f002:**
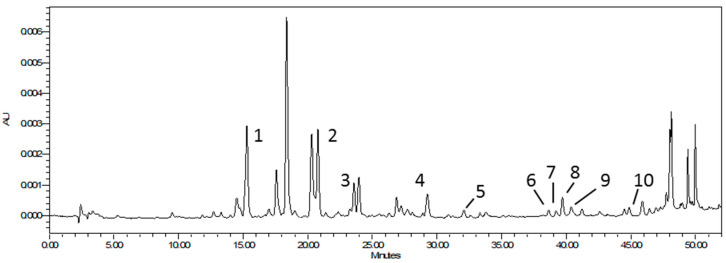
Chromatogram of black currant leaf extract with marked peaks of the identified compounds. The numbers 1–10 refer to the names of the compounds listed in Table 1.

**Figure 3 molecules-28-01749-f003:**
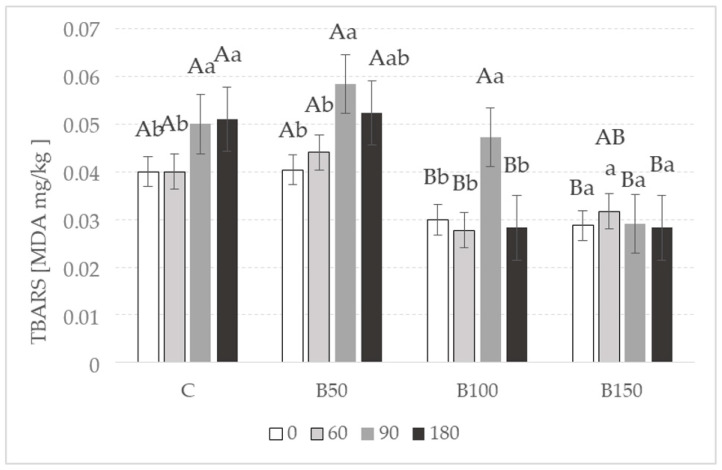
Amount of secondary lipid oxidation products (TBARS) in canned meat samples fortified with the extract of black currant leaves during 180 days of storage (4 °C). C—control; B50—black currant leaves 50 mg/kg; B100—black currant leaves 100 mg/kg; B150—black currant leaves 150 mg/kg. Means with different capital letters are significantly different (*p* < 0.05) on the day of the study. Means with different small letters are significantly different (*p* < 0.05) for a single variant during the storage time. Results are presented as mean ± SD.

**Table 1 molecules-28-01749-t001:** Chemical content and antioxidant activity of water extract obtained from black currant leaves.

Analyzed Parameters	Content
Extraction yield (%)	18.05 ± 0.15
Vitamin C (mg L-ascorbic acid/100 g)	3.11 ± 0.03
DPPH (EC50, μg/mL)	32.5 ± 0.21
ABTS (EC50, μg/mL)	11.1 ± 0.06
Total phenolic content (mg gallic acid/g)	100.5 ± 9.1
Flavonoid content (mg quercetin/g)	10.02 ± 0.1
Dihydroxycinnamic acid content (mg chlorogenic acid/g)	53.54 ± 2.85
Main phenolic compounds (HPLC) (mg/g):	
1. Caffeoyl malic acid	0.335 ± 0.005
2. Chlorogenic acid	5.273 ± 0.015
3. Caffeic acid	3.391 ± 0.021
4. Coumaric acid	0.264 ± 0.004
5. Ferulic acid	0.297 ± 0.025
6. Rutin	0.006 ± 0.001
7. Quercetin 3-*O*-glucoside	0.009 ± 0.001
8. Luteolin-7-*O*-rhamnoside	0.024 ± 0.002
9. Apigenin-7-*O*-glucoside	0.017 ± 0.001
10. Luteolin-7-*O*-glucoside	0.018 ± 0.001

**Table 2 molecules-28-01749-t002:** Minimum inhibitory concentration (mg/mL) of black currant leaf extract.

Bacterial Strain	MIC	Bacterial Strain	MIC
*Bacillus cereus* ATCC 11778	5	*Listeria monocytogenes* ATCC 15313	-
*Bacillus subtilis* ATCC 6633	5	*L. monocytogenes* ATCC 19111	5
*Clostridium sporogenes* ATCC 11437	-	*L. monocytogenes* ATCC 7644	5
*Enterococcus faecalis* ATCC 51229	-	*L. monocytogenes* IFM 1011	5
*Escherichia coli* ATCC 10536	-	*Salmonella enterica* ATCC 29631	5
*E. coli* ATCC 25922	-	*Salmonella* Hofit IFM 2318	-
Starter culture for meat fermentation M892	-	*Staphylococcus aureus* 4.4	2.5
*Lactobacillus plantarum* 299v	-	*S. aureus* 4538	1.5
*Lacticaseibacillus rhamnosus* ŁOCK 0900	-	*S. aureus* ATCC 25923	2.5
*Listeria innocua* ATCC 33090	-	*S. aureus* ATCC 6535	5.0

**Table 3 molecules-28-01749-t003:** The color properties (CIE L*a*b* system) and amount of meat pigment (nitrosohemochrome) in canned meat samples with a reduced amount of sodium nitrite and fortified with the extract of black currant leaves.

Parameter	Sample	Storage Time (Days)			
		1	60	90	180
L*	C	63.50 ± 3.01 ^Aa^	62.35 ± 2.98 ^Aa^	62.15 ± 1.60 ^Aa^	62.80 ± 2.58 ^Aa^
	B50	63.55 ± 3.29 ^Aa^	62.23 ± 3.02 ^Aa^	62.12 ± 1.64 ^Aa^	62.89 ± 2.63 ^Aa^
	B100	62.37 ± 2.94 ^Aa^	64.02 ± 2.57 ^Aa^	62.34 ± 2.48 ^Aa^	78.84 ± 2.91 ^Bb^
	B150	62.01 ± 3.06 ^Aa^	63.91 ± 3.30 ^Aa^	66.01 ± 2.36 ^Ba^	63.32 ± 2.24 ^Aa^
a*	C	9.68 ± 1.15 ^Ba^	10.59 ± 1.05 ^Bb^	10.71 ± 0.50 ^ABb^	9.90 ± 1.15 ^Ba^
	B50	9.72 ± 1.17 ^Ba^	10.68 ± 1.04 ^Bb^	10.78 ± 0.55 ^ABb^	9.95 ± 1.00 ^Ba^
	B100	10.13 ± 1.17 ^Aa^	10.01 ± 0.93 ^Aa^	10.73 ± 1.21 ^Aa^	12.38 ± 0.97 ^Ab^
	B150	10.29 ± 0,97 ^Ab^	10.11 ± 1.07 ^Bb^	9.45 ± 0.91 ^Bab^	9.83 ± 0.77 ^Ba^
b*	C	7.60 ± 0.70 ^Ab^	8.10 ± 0.57 ^Aab^	8.79 ± 0.62 ^Aa^	8.77 ± 0.73 ^Bb^
	B50	7.59 ± 0.68 ^Ab^	8.17 ± 0.65 ^Ab^	8.86 ± 0.44 ^Aa^	8.77 ± 0.84 ^Ba^
	B100	8.36 ± 0.65 ^Ab^	8.70 ± 0.56 ^Ab^	9.23 ± 0.64 ^Ab^	13.23 ± 0.54 ^Aa^
	B150	8.17 ± 0.76 ^Aa^	8.31 ± 0.58 ^Aa^	8.69 ± 0.58 ^Bab^	8.87 ± 0.73 ^Bb^
Nitrosohemochrome	C	21.95 ± 0.77 ^Ba^	24.37 ± 1.76 ^Ba^	21.98 ± 1.49 ^Ba^	22.51 ± 2.39 ^Aa^
	B50	22.73 ± 0.97 ^Ba^	25.59 ± 1.83 ^Ba^	22.51 ± 1.43 ^Ba^	22.48 ± 2.72 ^Aa^
	B100	20.26 ± 1.37 ^Aa^	29.16 ± 1.62 ^Cc^	19.40 ± 1.55 ^Aa^	22.67 ± 1.26 ^Ab^
	B150	27.84 ± 2.4 ^Cb^	23.54 ± 1.72 ^Aa^	24.24 ± 1.38 ^Cb^	22.00 ± 2.85 ^Aa^

C—control; B50—black currant leaves 50 mg/kg; B100—black currant leaves 100 mg/kg; B150—black currant leaves 150 mg/kg. Means with different capital letters are significantly different (*p* < 0.05) in the same column. Means with different small letters are significantly different (*p* < 0.05) in the same row. Results are presented as mean ± SD.

**Table 4 molecules-28-01749-t004:** Antioxidant abilities of canned meat fortified with the extract from black currant leaves during 180 days of storage (4 °C).

Parameter	Sample	Storage Time (Days)			
		1	60	90	180
ABTS^•+^ [mg_TROLOX_/mL]	C	3.90 ± 0.42 ^Aa^	3.20 ± 0.29 ^Bb^	2.71 ± 0.31 ^Bc^	3.65 ± 0.20 ^ABa^
	B50	3.95 ± 0.61 ^Aa^	3.22 ± 0.35 ^Bb^	2.67 ± 0.21 ^Bc^	3.71 ± 0.22 ^ABa^
	B100	2.37 ± 0.54 ^Bc^	4.42 ± 0.44 ^Aa^	4.02 ± 0.39 ^Aa^	3.61 ± 0.43 ^Bab^
	B150	3.49 ± 0.31 ^Ab^	3.91 ± 0.63 ^Aab^	3.76 ± 0.23 ^Aab^	4.04 ± 0.26 ^ABa^
DPPH [mg_TROLOX_/mL]	C	0.019 ± 0.001 ^Aa^	0.015 ± 0.001 ^Bb^	0.015 ± 0.001 ^Bb^	0.013 ± 0.001 ^Bc^
	B50	0.018 ± 0.001 ^Aa^	0.015 ± 0.001 ^Bb^	0.015 ± 0.001 ^Bb^	0.013 ± 0.001 ^Bc^
	B100	0.018 ± 0.001 ^Ab^	0.02 ± 0.000 ^Aa^	0.015 ± 0.001 ^Bc^	0.014 ± 0.002 ^Ad^
	B150	0.018 ± 0.001 ^Ab^	0.02 ± 0.001 ^Aa^	0.017 ± 0.001 ^Ab^	0.013 ± 0.001 ^Bc^
FRAP [A_700 nm_]	C	1.67 ± 0.08 ^Ba^	1.40 ± 0.04 ^Bc^	1.49 ± 0.10 ^Cb^	1.70 ± 0.04 ^Ba^
	B50	1.68 ± 0.09 ^Ba^	1.38 ± 0.05 ^Bc^	1.52 ± 0.11 ^Cb^	1.74 ± 0.04 ^Ba^
	B100	1.64 ± 0.16 ^Bb^	1.91 ± 0.08 ^Aa^	1.71 ± 0.03 ^Bb^	1.73 ± 0.02 ^Bb^
	B150	1.84 ± 0.07 ^Ab^	1.87 ± 0.07 ^Ab^	1.91 ± 0.11 ^Aab^	2.00 ± 0.05 ^Aa^

C—control; B50—black currant leaves—50 mg/kg; B100—black currant leaves—100 mg/kg; B150—black currant leaves—150 mg/kg. Means with different capital letters are significantly different (*p* < 0.05) in the same column. Means with different small letters are significantly different (*p* < 0.05) in the same row. Results are presented as mean ± SD.

## Data Availability

Not applicable.

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
