# Peer review of "Reduction of Nitrite in Canned Pork through the Application of Black Currant (*Ribes nigrum* L.) Leaves Extract"

_molecules, 2023, doi:10.3390/molecules28041749_

Round 1
Reviewer 1 Report
The manuscript (Reduction of nitrite in canned pork through the application of 2 Ribes nigrum L.) was involved in the replacement of nitrite in the canned pork with plant extract. After reviewing MS, the reviewer has found several mistakes in the text and missing important analyses require in the storage study. The current form of MS is not publishable in a well-reputed journal.
The detailed comments are given below.
The author should divide the manuscript into headings and subheadings, for example, Nitrosohemochrome, lipid oxidation, antioxidant activities, etc. should be discussed under their name.
Abstract: It majorly consists of an introduction and methodology; important results should be mentioned in it.
Scientific names: It should be in italics but throughout the MS, authors have used non-italic fonts, e.g. L40-4, 60-67, and many more.
Section 2.1: Did the authors optimize the extraction conditions of the leaf extract?? How can you assure that those mentioned conditions were the best ones?
Section 2.2.1: Why only radical scavenging activities were determined, how about other assays such as metal chelating activity, ORAC??
L154: AlCl3? please check.
L168: M or mol/L, units’ style should be consistent.
-For the plant name authors have shifted too many times between black currant and R. nigrum, which can be confusing. Therefore, it should be defined in the first place and then use only one name throughout the MS.
Section 2.5.5. Authors have determined only pathogenic bacteria contents, how about, total viable count, psychrophilic bacterial count, and other spoilage bacterial count? That bacterial load plays an important role in the color and lipid oxidation of the meat.
Table 3: should be nearby its first citation place, probably on page 10.
L382: Nitrosohemochrome content, should be explained in a separate section.
-Myoglobin content such as oxymyoglobin, metmyoglobin, etc. plays an important role in the color of meat, which is missing in the current study.
L382, 390: Mb-NO or NO-Mb, please be consistent and its full form should be given in the first place.
L402: NaNO2; please check
In Table 3, a* content was increased suddenly at day 180 of storage, why? It should be discussed in the text.
L441: During the time, What time??
Check the units for DPPH, ABTS, and FRAP, and increasing activities should be explained. Moreover, the discussion part is very weak, the authors have mentioned only results but the reason behind them is not enough to support their statements, for example in L504-506.
In Figure 3: Why there is an abrupt increase in TBARS in sample B100 on day 150 and then a sudden decrease on day 180??
-What is the shelf-life of samples, 180 days, or samples can be stored for a longer period??
Author Response
The manuscript (Reduction of nitrite in canned pork through the application of Ribes nigrum L.) was involved in the replacement of nitrite in the canned pork with plant extract. After reviewing MS, the reviewer has found several mistakes in the text and missing important analyses require in the storage study. The current form of MS is not publishable in a well-reputed journal.
Suggestion 1. The author should divide the manuscript into headings and subheadings, for example, Nitrosohemochrome, lipid oxidation, antioxidant activities, etc. should be discussed under their name.
Thank you for this suggestion. Indeed, the introduction of additional titles and subtitles in the manuscript has organized the data presented. Thanks to this, the reception of the content is logical and easier for the reader. So we re-examined the text and added subheadings as suggested. We made all the changes directly in the text.
Suggestion 2: Abstract: It majorly consists of an introduction and methodology; important results should be mentioned in it.
Thank you for this suggestion. Indeed, the version of the abstract presented earlier did not accurately reflect our results. We therefore fully agree that there was a need to improve this part of the manuscript. Therefore, we re-analyzed the abstract and corrected it, pointing out the most important observations. We now see that this treatment was necessary for a better understanding of our research. So thank you again for this tip.
Suggestion 3: Scientific names: It should be in italics but throughout the MS, authors have used non-italic fonts, e.g. L40-4, 60-67, and many more.
Thank you for pointing out the incorrect function of writing scientific names. Indeed, this is our mistake for which we apologize. It was due to our oversight. Therefore, we extracted and traced the entire text once again and applied all changes directly to the revised version of the manuscript.
Suggestion 4: Section 2.1: Did the authors optimize the extraction conditions of the leaf extract?? How can you assure that those mentioned conditions were the best ones?
Thank you for this this question. Of course, extraction conditions optimization analyzes were carried out in our lab. In these studies, the type of solvent, temperature and extraction time were taken into account, as well as the process was assisted, including ultrasound. In the discussed studies, we limited ourselves to solvents approved for contact with food, i.e. ethanol and water. Finally, water was selected as the optimal solvent, and ultrasonic extraction for 10 minutes was used as the supporting agent. The studies in question have been published (in Polish):
- Staszowska-Karkut M., Materska M., Kulik B., Waraczewski R.: Określenie wpływu rodzaju wody na potencjał antyoksydacyjny naparów z wybranych roślin ziołowych. Rośliny- przegląd wybranych zagadnień. Wydawnictwo Naukowe TYGIEL sp. Z o. o., Lublin 2016, str. 175-184.
- Staszowska-Karkut M., Materska M., Chilczuk B., Pabich M., Sachadyn –Król M.: Wpływ rodzaju rozpuszczalnika na aktywność przeciwrodnikową ekstraktów ziołowych. Żywność dla przyszłości, XLIII Sesja Naukowa Nauk o Żywności i Biotechnologii, Wrocław 2017, str. 221-228.
- Chilczuk B., Materska M., Staszowska-Karkut M., Kulik B., Stępnikowska A.: Porównanie metod ekstrakcyjnych przy przygotowywaniu preparatów roślinnych o korzystnych właściwościach prozdrowotnych. Żywność – tradycja i nowoczesność- Prozdrowotne właściwości żywności, aspekty żywieniowe i technologiczne. Red. M. Karwowska, I. Jackowska. Lublin 24-25.05.2018. str. 17-28
Thank you again for this remark.
Suggestion 5: Section 2.2.1: Why only radical scavenging activities were determined, how about other assays such as metal chelating activity, ORAC??
Due to the large number of studies carried out for the pure extract and food samples with its addition, we had to choose those that would best characterize its composition and activity. When choosing the method for determining the chemical activity of the extract, we focused on determining its activity spectrum in the hydrophilic system, these are determinations with the ABTS cation radical and the lipophilic system, which were performed for the DPPH radical. These are also the most frequently chosen methods of analyzing the chemical activity of plant extracts, thanks to which the obtained results can be compared with the studies of other authors. I would like to thank the Reviewer for this remark and in the future we will expand the spectrum of research with other research methods mentioned above.
Suggestion 6: L154: AlCl3? please check.
Thank you for pointing out our mistake at the stage of editing the text for review. Of course, this should not have happened, for which we apologize. It was checked and corrected as : AlCl3. We corrected the notation directly in the text.
Suggestion 7: L168: M or mol/L, units’ style should be consistent.
Thank you for this remark. Of course, the form M instead of mol/L is correct, as noted in the revised version of the manuscript.
Suggestion 8: For the plant name authors have shifted too many times between black currant and R. nigrum, which can be confusing. Therefore, it should be defined in the first place and then use only one name throughout the MS.
Thank you for your suggestion. Indeed, the unification of the name used introduces order in the text, which makes it more accessible to future readers. Therefore, we fully agree with the reviewer's opinion. We re-examined the text and decided that black currant would be more appropriate. We corrected all changes directly in the text. Thank you again for this tip.
Suggestion 9: Section 2.5.5. Authors have determined only pathogenic bacteria contents, how about, total viable count, psychrophilic bacterial count, and other spoilage bacterial count? That bacterial load plays an important role in the color and lipid oxidation of the meat.
Thank you for this suggestion. We would like to clarify that we have limited our microbiological tests to only microorganisms important from the point of view of product safety. We treated them as model microorganisms, particularly dangerous for consumers. The negative results of the assessment of the microbiological quality of our sterilized canned meat products would encourage us to extend the analyzes with more detailed tests. In addition, the sterilization process was carried out correctly, which effectively inhibited the growth of other spoilage microorganisms.
Suggestion 10: Table 3: should be nearby its first citation place, probably on page 10.
Thank you for this attention. We moved Table 3 .to a more appropriate place in the text (on the page 11), without disturbing the graphical presentation of the results.
Suggestion 11: L382: Nitrosohemochrome content, should be explained in a separate section.
Thank you for this suggestion. We separated the section on color parameters (L*, a*, b*) from the nitrosochemochrome content by separating them with Table 3 instead of creating a separate section. However, these parameters are quite interrelated, and they were used together when confronting our research results with the observations of other authors described in the literature.
Suggestion 12: Myoglobin content such as oxymyoglobin, metmyoglobin, etc. plays an important role in the color of meat, which is missing in the current study.
Thank you for this suggestion. Indeed, we did not analyze either oxymyoglobin or metmyoglobin content in this study. Undoubtedly, these parameters could be used as indicators of color change in this study, so we will undoubtedly remember this suggestion and implement it when designing the next study. Once again, thank you very much for this tip, which in the future may contribute to a better understanding of physicochemical changes in the meat product, especially affecting the color.
Suggestion 13: L382, 390: Mb-NO or NO-Mb, please be consistent and its full form should be given in the first place.
Thank you for pointing out our mistake. It was not intentional, the error was probably made during language correction. Of course, the correct form is NO-Mb referring to nitrosochemochrome (a denatured form of nitrosohemoglobin). Changes have been made in the new version of the manuscript
Suggestion 14: L402: NaNO2; please check
It should be agreed that the number (2) should be described with a subscript. We fixed our mistake. Thank you for pointing it out.
Suggestion 15: In Table 3, a* content was increased suddenly at day 180 of storage, why? It should be discussed in the text.
Thank you for this question. The addition of a plant extract may affect, among others, the parameters of the meat product, such as, for example, pH, but also other properties. These, in turn, may inhibit or intensify biochemical changes in the meat product, which is the cause of the described trend in color parameters. However, to completely confirm our assumptions, we did not provide enough evidence in this study. Now that we have observed the effect of the extract content on the color parameters, we want to explain this phenomenon in detail in the future.
Suggestion 16: L441: During the time, What time??
Thank you for this remark. Of course, it's about storage periot. It was an unfortunate term due to language differences. This should not have happened, for which we apologize to the Reviewer. We would like to thank you once again for the effort and time devoted to such a thorough and thorough review of our manuscript. Of course, the text was re-analysed and we improved the form of the description.
Suggestion 17: Check the units for DPPH, ABTS, and FRAP, and increasing activities should be explained. Moreover, the discussion part is very weak, the authors have mentioned only results but the reason behind them is not enough to support their statements, for example in L504-506.
Thank you for this suggestion. After re-examining the text, we indeed see that the discussion of the results regarding antioxidant activity was not enough. Therefore, we have carefully analyzed this text and supplemented our descriptions. We made all changes directly in the manuscript. In addition, the bibliography was supplemented with the following entries:
- Anese, M., Manzocco, L., Nicoli, M. C., & Lerici, C. R. (1999). Antioxidant properties of tomato juice as affected by heating. Journal of the Science of Food and Agriculture, 79(5), 750-754.
- Tepe, B., Sokmen, M., Akpulat, H. A., Daferera, D., Polissiou, M., & Sokmen, A. (2005). Antioxidative activity of the essential oils of Thymus sipyleus subsp. sipyleus var. sipyleus and Thymus sipyleus subsp. sipyleus var. rosulans. Journal of Food Engineering, 66(4), 447-454.
- Guillen-Sans, R., & Guzman-Chozas, M. (1998). The thiobarbituric acid (TBA) reaction in foods: a review. Critical reviews in food science and nutrition, 38(4), 315-350.
- Wolosiak, R., Druzynska, B., Piecyk, M., Worobiej, E., Majewska, E., & Lewicki, P. P. (2011). Influence of industrial sterilisation, freezing and steam cooking on antioxidant properties of green peas and string beans. International journal of food science & technology, 46(1), 93-100.
- Ferysiuk, K., Wójciak, K. M., & Kęska, P. (2022). Effect of willow herb (Epilobium angustifolium L.) extract addition to canned meat with reduced amount of nitrite on the antioxidant and other activities of peptides. Food & Function, 13(6), 3526-3539.
- Mira, L., Tereza Fernandez, M., Santos, M., Rocha, R., Helena Florêncio, M., & Jennings, K. R. (2002). Interactions of flavonoids with iron and copper ions: a mechanism for their antioxidant activity. Free radical research, 36(11), 1199-1208.
Suggestion 18: In Figure 3: Why there is an abrupt increase in TBARS in sample B100 on day 150 and then a sudden decrease on day 180??
The TBARS test is a parameter to assess the degree of fat peroxidation. Fat oxidation products capable of reacting with thiobarbituric acid testify to the advancement of unfavorable transformations. A higher level of TBARS in the initial periods of storage proves progressive changes in fat oxidation. In the last study period, there was a decrease in TBARS values, which was best seen between day 90 and day 180 of the analysis in samples B100 and B150. This observation should not be understood unambiguously as a slowdown of oxidative changes in meat products. As a result of transformations caused by oxidative factors, the products of fat oxidation were chemically transformed into smaller compounds that can no longer be detected by the TBARS test.
Suggestion 19: What is the shelf-life of samples, 180 days, or samples can be stored for a longer period??
Our research was completed on day 180 of the analysis. After this time, we received a product with satisfactory quality parameters. In particular, the microbiological quality (based on selected microorganisms that are particularly dangerous for the consumer) and the low level of nitrosamines prove a high level of product safety. With high probability, we can say that our canned meat product could be stored for much longer, however, as part of this study, we did not analyze further biochemical changes in subsequent periods of time.
We would like to thank you again very much for your time and such an insightful review. All comments are very important to us, so we have implemented them. We trust that the new version of the manuscript is of high scientific value and is suitable for publication in the Molecules journal.
Reviewer 2 Report
The authors used water extract of R. nigrum as additive for canned meat to reduce the amount of nitrite required for conservation. The information is well presented and the results are relevant to the subject.
There are very few observations, mainly focused on improving the discussion of the results.
All scientific names of plants and microorganisms should be written in italics, please revise all the text
Lines 107 and 109: Freezing temperatures should have negative values ( -80°C)
Lines 325-328: The standard deviation of the EC50 values should be added to the text.
Table 1: Units should be added to the table.
Table 2: Why is the MIC reported as percentage? I recommend reporting the MIC values as ug/mL
Table 3: Its interesting that color properties change with B100 in comparison with B50 and B150. Is there an explanation or discussion of this results? It doesn’t seem to be cause by lack or excess of plant extract.
Lines 436-474: Although the results are well explained and presented, I recommend discussing your own results and give a possible explanation to the behavior of the results.
Lines 501-503: please check the writing
Author Response
The authors used water extract of R. nigrum as additive for canned meat to reduce the amount of nitrite required for conservation. The information is well presented and the results are relevant to the subject.
There are very few observations, mainly focused on improving the discussion of the results.
Suggestion 1: All scientific names of plants and microorganisms should be written in italics, please revise all the text
Thank you for this suggestion. Of course, we agree with this opinion. This was our oversight, which should not be the case when submitting a text for review. Therefore, we re-examined the entire text and made the appropriate changes directly in the revised version of the manuscript.
Suggestion 2: Lines 107 and 109: Freezing temperatures should have negative values ( -80°C)
Thank you for pointing out our mistake. Of course, we agree that the record was wrong. We stored our samples in deep-freezing conditions (-80). Appropriate changes have been made in the text.
Suggestion 3: Lines 325-328: The standard deviation of the EC50 values should be added to the text.
We agree with the Reviewer's opinion that standard deviations should be included in the text of the manuscript. This procedure facilitates the presentation of the results and their proper interpretation. Thus, the standard deviation was inserted into the text
Suggestion 4: Table 1: Units should be added to the table.
Thank you for this remake. Table 1 was completed in the second column, as present in revised version of the manuscript
Suggestion 5: Table 2: Why is the MIC reported as percentage? I recommend reporting the MIC values as ug/mL
Thank you for this tip. Indeed, the presentation of results as g/ml is more appropriate for high-quality research papers. Therefore, we re-edited the text and made appropriate changes in Table 2.
Suggestion 6: Table 3: Its interesting that color properties change with B100 in comparison with B50 and B150. Is there an explanation or discussion of this results? It doesn’t seem to be cause by lack or excess of plant extract.
Thank you for this question. The addition of a plant extract may affect, among others, the parameters of the meat product, such as, for example, pH, but also other properties. These, in turn, may inhibit or intensify biochemical changes in the meat product, which is the cause of the described trend in color parameters. However, to completely confirm our assumptions, we did not provide enough evidence in this study. Now that we have observed the effect of the extract content on the color parameters, we want to explain this phenomenon in detail in the future.
Suggestion 7: Lines 436-474: Although the results are well explained and presented, I recommend discussing your own results and give a possible explanation to the behavior of the results.
- Anese, M., Manzocco, L., Nicoli, M. C., & Lerici, C. R. (1999). Antioxidant properties of tomato juice as affected by heating. Journal of the Science of Food and Agriculture, 79(5), 750-754.
- Tepe, B., Sokmen, M., Akpulat, H. A., Daferera, D., Polissiou, M., & Sokmen, A. (2005). Antioxidative activity of the essential oils of Thymus sipyleus subsp. sipyleus var. sipyleus and Thymus sipyleus subsp. sipyleus var. rosulans. Journal of Food Engineering, 66(4), 447-454.
- Guillen-Sans, R., & Guzman-Chozas, M. (1998). The thiobarbituric acid (TBA) reaction in foods: a review. Critical reviews in food science and nutrition, 38(4), 315-350.
- Wolosiak, R., Druzynska, B., Piecyk, M., Worobiej, E., Majewska, E., & Lewicki, P. P. (2011). Influence of industrial sterilisation, freezing and steam cooking on antioxidant properties of green peas and string beans. International journal of food science & technology, 46(1), 93-100.
- Ferysiuk, K., Wójciak, K. M., & Kęska, P. (2022). Effect of willow herb (Epilobium angustifolium L.) extract addition to canned meat with reduced amount of nitrite on the antioxidant and other activities of peptides. Food & Function, 13(6), 3526-3539.
- Mira, L., Tereza Fernandez, M., Santos, M., Rocha, R., Helena Florêncio, M., & Jennings, K. R. (2002). Interactions of flavonoids with iron and copper ions: a mechanism for their antioxidant activity. Free radical research, 36(11), 1199-1208.
Suggestion 8: Lines 501-503: please check the writing
The text was corrected as suggested by the Reviewer. Thank you for this tip.
We would like to thank you again very much for your time and such an insightful review. All comments are very important to us, so we have implemented them. We trust that the new version of the manuscript is of high scientific value and is suitable for publication in the Molecules journal.
Reviewer 3 Report
The study entitled as “Reduction of nitrite in canned pork through the application of Ribes nigrum L.” has focused on the addition of lyophilized extracts that were obtained from black currant leaves to the canned meat. For this purpose canned meats were stored and monitored for 180 days, measuring many properties such as color, TBARS and antioxidative capacity.
My literature survey showed that blackcurrant leaf extract has not been used for its antioxidant and antimicrobial properties in canned meat. However, the title of the study does not give idea about the extract was obtained from the leaves of this material. Then, title should be revised.
Abstract should be improved, since it doesn’t give any numerical results and not able to reflect the whole concept of the study.
Introduction was well written giving the required information and previous studies, however objective paragraph does not reflect the main objective of the study well.
Materials and methods part successfully gives the details about the analyses. Results were given in appropriate tables and figures with the statistical tools applied. In some parts of the results, discussions should be improved (please find in uploaded pdf)
Please find my extra comments given in the Comments part of uploaded pdf file.

Author Response
The authors used water extract of R. nigrum as additive for canned meat to reduce the amount of nitrite required for conservation. The information is well presented and the results are relevant to the subject.
There are very few observations, mainly focused on improving the discussion of the results.
All scientific names of plants and microorganisms should be written in italics, please revise all the text
Suggestion 1: Title: better to mention about the use of its leaves extracts.
Thank you for this tip. We fully agree with the opinion of the Reviewer that the title of the manuscript should be clarified to better reflect the scope of our research. We therefore reworded it as "Reduction of nitrite in canned pork through the application of black currant (Ribes nigrum L.) leaves extract", as shown in the new revised version of the manuscript.
Suggestion 2. Introduction: lease revise the sentence, it gives the idea that canned pork has a.o., a.m. or color forming properties. Please mention about any toxic compound of R. nigrum leaves in somewhere appropriate.
Thank you for this suggestion, which is especially important to us. Introduction is part of the manuscript which should illuminate the context of the presented research. The version of the manuscript submitted for review did not accurately represent their scope. Therefore, we re-examined this part and made corrections in the last paragraph of the Introduction section.
Suggestion 3: Position2.2. Please give number of replicates for each type of analysis, in somewhere appropriate.
Thank you for pointing out our oversight, for which we are very sorry. This should not be the case when submitting a manuscript for peer review. As suggested by the reviewer, we included an annotation regarding the number of repetitions we carried out for each study. We mentioned that all measurements were carried out in triplicate. We have included a relevant description in the Statistical analysis subsection, which can be seen in the new, revised version of the manuscript.
Suggestion 4: It is not clear whether 50 mg or sodium nitrite was added to 1 kg of meat or 1 kg of gross can content. the same is valid for the leaf extract addition. please clarifiy.
Thank you for this question. It should be clarified that both: the addition of sodium nitrate and herbs extract have been added to the raw meat stuffing (meat). The previous record was probably ambiguous and could be misleading. Therefore, in the new, revised version of the manuscript, we clarified that it concerns meat, and not the finished product (gross content of the can).
Suggestion 5: MDA please give its open form
Thank you for pointing out our oversight. We fully agree that all abbreviations used in the text should be explained in order to fully understand the content. Therefore, directly in the text of the manuscript, we explained that MDA should be understood as malondialdehyde.
Suggestion 6: Rich - this is not an objective word
Thank you for this tip. Indeed, the word "rich" is not the right word for high-quality research papers. Therefore, we re-examined this part of the text and corrected it by removing redundant descriptions. Now that sentence reads "Chemical analysis revealed the complex chemical composition of the prepared extract (Table 1)"
Suggestion 7: Please clarify the form of BHT. was it a solution, if yes what was its concentration?
It should be clarified that BHT was in the form of a powder, and was prepared under the same conditions as those used in the antiradical activity tests. For the avoidance of doubt, we have completely removed this sentence.
Suggestion 8: Line 335: please clarify the difference. was the obtained value lower or higher, please compare.
Thank you for this suggestion. We fully agree with the Reviewer's opinion that this provision was not clear enough. The chemical content of the investigated dry black currant leaf extract differed from that reported by other authors both quantitatively and qualitatively. The highest concentration of chlorogenic acid was recorded in the tested extract, while in other studies the highest concentration of quercetin-3-O-glucoside was noted. This information has been added in the new version of the manuscript.
Suggestion 9: please give a title name to the seond column (Table 1).
Table 1 was completed in the second column
Suggestion 10: please avoid using P>0.05, since the decision is given with P<0.05 criterion even it is significant or not. please check throughout the text.
Thank you for this suggestion. It is especially important to us. Thanks to it, we can improve the scientific value of our manuscript. We re-analysed and re-edited the entire text and decided to use the P sign only for statistically significant differences (P<0.05). Thanks again for this tip
Suggestion 11: On the first day of storage, the addition of blackcurrant leaf extract did not differ the antioxidant properties of canned meat significantly compared to the control. this is an unexpected result and the discussion should be improved.
- Anese, M., Manzocco, L., Nicoli, M. C., & Lerici, C. R. (1999). Antioxidant properties of tomato juice as affected by heating. Journal of the Science of Food and Agriculture, 79(5), 750-754.
- Tepe, B., Sokmen, M., Akpulat, H. A., Daferera, D., Polissiou, M., & Sokmen, A. (2005). Antioxidative activity of the essential oils of Thymus sipyleus subsp. sipyleus var. sipyleus and Thymus sipyleus subsp. sipyleus var. rosulans. Journal of Food Engineering, 66(4), 447-454.
- Guillen-Sans, R., & Guzman-Chozas, M. (1998). The thiobarbituric acid (TBA) reaction in foods: a review. Critical reviews in food science and nutrition, 38(4), 315-350.
- Wolosiak, R., Druzynska, B., Piecyk, M., Worobiej, E., Majewska, E., & Lewicki, P. P. (2011). Influence of industrial sterilisation, freezing and steam cooking on antioxidant properties of green peas and string beans. International journal of food science & technology, 46(1), 93-100.
- Ferysiuk, K., Wójciak, K. M., & Kęska, P. (2022). Effect of willow herb (Epilobium angustifolium L.) extract addition to canned meat with reduced amount of nitrite on the antioxidant and other activities of peptides. Food & Function, 13(6), 3526-3539.
- Mira, L., Tereza Fernandez, M., Santos, M., Rocha, R., Helena Florêncio, M., & Jennings, K. R. (2002). Interactions of flavonoids with iron and copper ions: a mechanism for their antioxidant activity. Free radical research, 36(11), 1199-1208.
Suggestion 12: please cite the related table
Thank you for this suggestion. Of course, we corrected all text and filled in the missing table references (or we indicated that the data was not presented – subsection 3.2.3.). The changes are visible in the new version of the manuscript.
Suggestion 13: "strong" does not fit well with the obtained results, please revise.
Indeed, the wording we placed was not correct. That's why we changed the word "strong" to the word "high" in the context of antioxidant properties. We now see that this treatment was necessary and more appropriate for a manuscript published in a highly reputable journal.
We would like to thank you again very much for your time and such an insightful review. All comments are very important to us, so we have implemented them. We trust that the new version of the manuscript is of high scientific value and is suitable for publication in the Molecules journal.
Round 2
Reviewer 1 Report
The manuscript has improved as compared to the previous version. However, authors should consider that TVC, PBC and spoilage bacterial contents are theimportant parameters to conclude the shelf-life of any product.
Reviewer 3 Report
Can be accepted as it is